# Determining Heat Stress Effects of Multiple Genetic Traits in Tropical Dairy Cattle Using Single-Step Genomic BLUP

**DOI:** 10.3390/vetsci9020066

**Published:** 2022-02-03

**Authors:** Piriyaporn Sungkhapreecha, Vibuntita Chankitisakul, Monchai Duangjinda, Sayan Buaban, Wuttigrai Boonkum

**Affiliations:** 1Department of Animal Science, Faculty of Agriculture, Khon Kaen University, Khon Kaen 40002, Thailand; pat_sungkhapreecha@hotmail.com (P.S.); vibuch@kku.ac.th (V.C.); monchai@kku.ac.th (M.D.); 2Network Center for Animal Breeding and Omics Research, Khon Kaen University, Khon Kaen 40002, Thailand; 3Bureau of Animal Husbandry and Genetic Improvement, Department of Livestock Development, Pathum Thani 12000, Thailand; buaban_ai@hotmail.com

**Keywords:** heat tolerance, milk fat-to-protein ratio, milk yield, somatic cell score, Thai-Holstein crossbred

## Abstract

Heat stress is becoming a significant problem in dairy farming, especially in tropical countries, making accurate genetic selection for heat tolerance a priority. This study investigated the effect of heat stress manifestation on genetics for milk yield, milk quality, and dairy health traits with and without genomic information using single-step genomic best linear unbiased prediction (ssGBLUP) and BLUP in Thai−Holstein crossbred cows. The dataset contained 104,150 test-day records from the first lactation of 15,380 Thai−Holstein crossbred cows. A multiple-trait random regression test-day model on a temperature−humidity index (THI) function was used to estimate the genetic parameters and genetic values. Heat stress started at a THI of 76, and the heritability estimates ranged from moderate to low. The genetic correlation between those traits and heat stress in both BLUP methods was negative. The accuracy of genomic predictions in the ssGBLUP method was higher than the BLUP method. In conclusion, heat stress negatively impacted milk production, increased the somatic cell score, and disrupted the energy balance. Therefore, in dairy cattle genetic improvement programs, heat tolerance is an important trait. The new genetic evaluation method (ssGBLUP) should replace the traditional method (BLUP) for more accurate genetic selection.

## 1. Introduction

Against the background of a changing climate, heat stress significantly impacts the production, fertility, health, and welfare of livestock animals, including dairy cattle [1,2,3], in Thailand and in many other countries around the world [4,5,6,7,8]. In addition, increases in temperature and humidity have been affecting dairy cattle [1,2], leading to heat stress [3,9,10]. When subjected to heat stress, dairy cattle respond with reduced feed intake and rumination and increased water intake, as well as with a higher respiratory rate, increased sweating, more panting, and impeded energy balance [7,11,12]. Consequently, milk yield and reproductive performance are decreased and health deteriorates, sometimes even leading to death [5,13,14,15,16,17]. Currently, there are several approaches to reduce heat stress in dairy cattle, such as improving the environment and installing cooling devices, managing nutrition, and improving their genetics using crossbreeds between tropical and temperate dairy cattle [11,17,18,19,20]. However, these approaches have different efficiencies, and each method must be adjusted according to the changes in climatic conditions. Moreover, farmers still lack profound knowledge in this aspect, making it important to consider alternative methods that provide sustainable results.

In Thailand, it can be difficult to improve the environment and install cooling devices because most dairy farms are small-scale farms (>80%) [21], and there are labor and budget constraints [5]. Moreover, crossbreeding between *Bos taurus* (Holstein, Jersey, Brown Swiss, and Red Dene) and *Bos indicus* (Sahiwal, Red Sindhi, Brahman, and Thai Native) is widespread; most crossbred dairy cattle (>87.5% Holstein genetics) are crossbreeds between Holstein and Sahiwal or Thai Native breeds. These two breeds are generally crossed to obtain the high milk yield from *Taurine* and the tolerance to heat, ticks, and tropical diseases from *Zebu* [22,23]. However, grading up with Holstein to improve milk yield has been applied without any emphasis on heat stress, resulting in yield losses.

One possible way to sustainably resolve the issue is genetic selection for dairy cattle with heat tolerance. The most used measure of the heat stress level is the temperature-humidity index (THI), in combination with the selection of superior dairy cattle breeders [24,25]. The first study on Thai dairy cattle genetic evaluation was published in 2011 [5]. The authors used the conventional method to study the effects of heat stress on milk production, which is an economically major trait producing benefits to dairy farmers. In addition, a high somatic cell score (SCS) is also related to mastitis, a major problem in the Thai dairy population, because it results in decreased milk synthesis [26]. Furthermore, the milk fat to protein ratio (FPR) is an easily measurable trait and can be used as an indicator to evaluate negative energy balance (NEB) and ketosis, which is also related to the decreasing milk yield [27]. Therefore, those traits are interesting to study together with heat stress in order to improve the genetic of production traits. However, the accuracy of the genetic approach for predictions is another concern.

Genomic selection is a new method of genetic evaluation that has been gaining increased attention and is widely used in several countries such as the United States, Canada, and Australia [8,28,29,30], due to its many advantages compared to conventional methods: (1) animals can be selected at a very early age, while selection by estimated breeding value (EBV) must be done after animals start producing, and their performance records would be used for genetic assessment and selection [31], and (2) genomic selection reduces the incidence of inbreeding [32], decreases generation interval, and increases genetic progress [31], especially low heritability traits such as fertility and dairy health [33].

Misztal et al. [34], Legarra et al. [35], and Christensen and Lund [36] proposed a single-step genomic best linear unbiased prediction method (ssGBLUP) for estimating genetics using pedigree, phenotypic, and genomic data in one step. It reduces bias and has a higher efficiency and accuracy [37,38,39]. To test the accuracy, Legarra and Reverter [40] preferred the linear regression (LR) method by comparing the accuracy of the genomic breeding value (GEBV) from partial and whole data. Holifield et al. [41] reported that the method was reliable and easy to calculate, and Bermann et al. [42] tested its accuracy in broilers, detecting a low bias and obtaining accurate results when comparing EBV and GEBV. With several advantages of the genomic approach, we hypothesized that the use of genomic information can improve the accuracy of estimated breeding values in the Thai−Holstein population. The objective of this study was to investigate the effect of heat stress manifestation on genetic parameters for milk yield, milk quality, and health with and without genomic information using ssGBLUP and BLUP in Thai−Holstein crossbred dairy cattle under hot and humid conditions.

## 2. Materials and Methods

### 2.1. Data

This studied was approved by the Institutional Animal Care and Use Committee of Khon Kaen University (no. IACUC-KKU-120/64). A total of 104,150 test-day records for milk yield (MY), somatic cell score (SCS), and milk fat-to-protein ratio (FPR) from the first lactation of 15,380 Thai−Holstein crossbred cows between 1999 and 2018 were provided by the dairy database of the Bureau of Biotechnology for Livestock Production, Department of Livestock Development, Thailand. The data recorded to be analyzed for genetic parameters and genetic values needed to have the following characteristics: (1) records taken 6 days before production or after 305 days in production were excluded, and only cows with at least five records were used in the analyses; (2) calving ages were restricted from 24 to 48 months; (3) information of herd-test-date combinations that had records less than 30 cows or daughters from less than three sires were excluded; and (4) cows were grouped by percentage of Holstein genes (breed group; BG) as follows: BG1 < 87.5%, BG2 from 87.5 to 93.6%, and BG3 > 93.7% [5]. A pedigree file was constructed by tracing back three generations of ancestors and included 33,799 individuals. The reference population of 882 animals was genotyped using the Illumina BovineSNP50 Bead Chip (Illumina Inc., San Diego, CA, USA). Quality control on animals and markers was performed according to the following parameters: minimum call rate equal to 90% and a minor allele frequency of each marker greater than 5%. Animals and markers that failed these quality control criteria were removed. All of the animals were retained, and a total of 43,288 SNP markers were retained for the genetic parameter estimation. The numbers of animals with genotypes, phenotypes, and pedigree records are presented in Table 1.

Climate data were obtained from the meteorological center closest to each dairy farm, based on the postal code (distance not more than 30 km). These data included daily temperature and relative humidity recorded every 3 h, which were used to calculate a temperature−humidity index (THI) based on the formula used by the National Oceanic and Atmospheric Administration [43]:THI=(1.8T+32)−(0.55−0.0055RH)(1.8T−26),
where T is the temperature in degrees celsius and RH is the relative humidity in percentage. The mean daily THI for the third day before each test day from the weather station closest to the farm was assigned as the THI for that test day, as suggested by Bohmanova et al. [44]. ${\mathrm{THI}}_{\mathrm{threshold}}$ was set to 76, as in Sungkhapreecha et al. [45]. A function (f) of THI was created:$$f\left(THI\right)=\{\begin{array}{cc} 0, & THI\leq {\mathrm{THI}}_{\mathrm{threshold}}\;\left(no\;heat\;stress\right)\\ \mathrm{THI}-{\mathrm{THI}}_{\mathrm{threshold}}, & \mathrm{THI}>{\mathrm{THI}}_{\mathrm{threshold}}\;\left(\mathrm{heat}\;\mathrm{stress}\right) \end{array}$$

### 2.2. Analyses and Computations

Genetic parameters and breeding values were estimated using the REMLF90 program [46] for a multiple-trait random regression test-day model on a THI function, as follows:$$y_{ijklmno}=hmy_{i}+fs_{j}+bgmim_{kl}+afc_{m}+bg_{k}\left[f\left(THI\right)\right]+a_{n}+a_{ht_{n}}\left[f\left(THI\right)\right]+p_{n}+p_{ht_{n}}\left[f\left(THI\right)\right]+e_{ijklmno},$$
where $y_{ijklmno}$ is the test-day milk yield (MY), somatic cell score (SCS), and milk fat-to-protein ratio (FPR) of cow o in herd-test month-test year (hmy) class *i* (i = 1 to 24,928), farm-calving season (fs) class *j* (j = 1 to 218), months in milk (mim) class *l* (*l* = 1 to 10) nested within breed group (bg) class *k* (*k* = 1 to 3), age at first calving (afc) class m (m = 1 to 7); $bg_{k}\left[f\left(THI\right)\right]$ is the slope (regression coefficient) of the decline of traits (MY, SCS, and FPR) per unit of change of the Holstein genetic level; a is the random additive genetic effect without consideration of heat tolerance (intercept); $a_{ht}$ is the random additive genetic effect for heat tolerance (slope); p is the random permanent environmental effect without consideration of heat stress (intercept); $p_{ht}$ is the random permanent environmental effect of heat tolerance (slope); e is the random residual effect; and f(THI) is a function of THI. The (co)variance structure is as follows:$$\mathrm{Var}\left[\begin{array}{c} a\\ \begin{array}{c} p\\ \begin{array}{c} h\\ e \end{array} \end{array} \end{array}\right]\left[\begin{array}{c} AxG_{0}\\ \begin{array}{c} 0\\ \begin{array}{c} 0\\ 0 \end{array} \end{array} \end{array}\begin{array}{c} 0\\ \begin{array}{c} IxP_{0}\\ \begin{array}{c} 0\\ 0 \end{array} \end{array} \end{array}\begin{array}{c} 0\\ \begin{array}{c} 0\\ \begin{array}{c} IxH_{0}\\ 0 \end{array} \end{array} \end{array}\begin{array}{c} 0\\ \begin{array}{c} 0\\ \begin{array}{c} 0\\ IxR_{0} \end{array} \end{array} \end{array}\right],$$
where A is the numerator relationship matrix; *G*_0_ and *P*_0_ are 6 × 6 matrices of (co)variances for additive (intercept and slope) and permanent environmental (intercept and slope) effects, respectively; *H*_0_ and *R*_0_ are the diagonal matrices of herd-month-year of test and residual variances corresponding to each trait; I is the identity matrix; and a, p, *h*, and *e* are defined as above.

Analyses were performed with and without genomic information, using single-step genomic BLUP (ssGBLUP) and conventional BLUP (BLUP) methods. In the ssGBLUP method, the inverse of the pedigree relationship matrix (A) in the mixed model equations is replaced by the inverse of the realized relationship matrix (H) [34,35,36,37], which is expressed as follows:$$H^{-1}=A^{-1}+\left[\begin{array}{cc} 0 & 0\\ 0 & G^{-1}-A_{22}^{-1} \end{array}\right],$$
where $A^{-1}$ is the inverse of a pedigree-based relationship matrix for all animals included in the analysis, $A_{22}^{-1}$ is the inverse of the pedigree-based relationship matrix for genotyped animals only, and $G^{-1}$ is the inverse of a genomic relationship matrix (G), constructed as in VanRaden [47]. Estimated breeding values were computed using the following equation [10]:$$\mathrm{BV}=a+f\left(\mathrm{THI}\right)*a_{ht},$$
where a is the breeding value without consideration of the heat stress effect (intercept), $a_{ht}$ v is the heat-tolerance breeding value (slope), and f(THI) is a function of the temperature humidity index.

### 2.3. Accuracy of Breeding Value Predictions

Statistics from the linear regression method (LR) [40] validate the intercept and slope estimated breeding values. A partial dataset (p) was constructed by removing the phenotypes of the bulls born between 1996 and 2011 (using only young bull candidates for selection), which were identified as focal individuals. The focal individuals did not have daughters within the data records (MY, SCS, and FPR) in the partial dataset, but in the whole dataset (w), bulls will have both bulls with daughter records and young bulls without daughter records. The average number of daughters with records for these focal individuals was four records. Two statistics for accuracies were used in this study and can be calculated as follows:

1. The ratio of accuracies is calculated as the correlation of EBV between the partial $\left({\hat{u}}_{p}\right)$ and whole $\left({\hat{u}}_{w}\right)$ dataset, using the following equation:$$\rho_{w,p}=\frac{cov\;\left({\hat{u}}_{w},{\hat{u}}_{p}\right)}{\sqrt{var\;({\hat{u}}_{w})var\;({\hat{u}}_{p})}}$$

2. The accuracy of genomic predictions was calculated as the covariance of EBV based on the partial and whole dataset, as follows:
$$\hat{Acc}\left(LR\right)=\sqrt{\frac{cov\left({\hat{u}}_{w},{\hat{u}}_{p}\right)}{\left(1-\bar{F}\right)\sigma_{a}^{2}}},$$
where $cov\;\left({\hat{u}}_{w},{\hat{u}}_{p}\right)$ is covariance between the estimated breeding values obtained with the whole dataset and the estimated breeding values obtained with the partial dataset; $\bar{F}$ is the average inbreeding coefficient in the validation population, and $\sigma_{a}^{2}$ is the additive genetic variance of the population.

## 3. Results

### 3.1. Effects of Heat Stress on Genetic Parameters

Based on Sungkhapreecha et al. [45], the threshold point of heat stress was found at a THI of 76. Thailand has a year-round THI in the range of range of 72 to 84, and on 250 days, the THI is greater than 76. This means that dairy cows raised in Thailand are affected by heat stress most of the year. The variance component estimates of MY, SCS, and FPR are presented in Table 2. The intercept values (additive genetic effect without consideration of heat tolerance) for MY, SCS, and FPR were 5.753, 0.152, and 0.007 for the BLUP method, respectively, and 5.650, 0.204, and 0.007 for the ssGBLUP method, respectively. The slopes (additive genetic effect for heat tolerance) for MY, SCS, and FPR were 0.094, 0.035, and 0.001 for the BLUP method, respectively, and 0.015, 0.005, and 0.001 for the ssGBLUP method, respectively. The covariance between intercept and slope (additive genetic effect) was negative, ranging from −0.116 to −0.001 for all of the traits and all methods. Estimates of heritability ranged from moderate to low, with values of 0.334, 0.049, and 0.060 for MY, SCS, and FPR traits, respectively, in the BLUP method. Heritability values in the ssGBLUP method were 0.344, 0.087, and 0.061 for MY, SCS, and FPR traits, respectively. Estimates for variance components of permanent environmental effects $\left(\sigma_{p}^{2}\right)$ were higher than those for genetic effects, with values of 6.208, 0.707, and 0.023 for MY, SCS, and FPR traits, respectively, in the BLUP method. The $\sigma_{p}^{2}$ values for the ssGBLUP method were 6.481, 0.754, and 0.025 for MY, SCS, and FPR traits, respectively (Table 2).

### 3.2. Genetic Correlation

Correlations between additive effects with and without heat stress considered (Table 2) were negative for MY and FPR traits, and ranged from −0.158 to −0.378 for the BLUP method and −0.368 to −0.438 for the ssGBLUP method. On the other hand, correlations between additive effects with and without heat stress considered were positive for SCS in both methods. Correlations between permanent environmental effects with and without heat stress considered were also negative and ranged from −0.119 to −0.204 for the BLUP method and −0.339 to −0.486 for the ssGBLUP method. The genetic and phenotypic correlation estimates of MY, SCS, and FPR traits for each method are given in Table 3. Genetic correlations between MY and SCS, and between MY and FPR were 0.06 and −0.21; the correlation between SCS and FPR was 0.06 for the BLUP method, whereas for the ssGBLUP method, these values were 0.09, −0.18, and 0.01, respectively.

### 3.3. Accuracy of Genetic Predictions

Table 4 shows the accuracy results using the linear regression method. The ssGBLUP method had a 32–96% higher accuracy for genetic prediction than the BLUP method. The ratios of accuracies for the BLUP method were 0.24, 0.37, and 0.26 in MY, SCS, and FPR; for the ssGBLUP method, these values were 0.37, 0.49, and 0.38, respectively. These results according to the accuracies were 0.20 to 0.25 for the BLUP method and 0.33 to 0.40 for the ssGBLUP method.

### 3.4. Rate of Decline in MY, SCS, and FPR Traits

The reduction in trait values due to heat stress at a THI of 76 in relation to the blood levels of Holstein cows is shown in Table 5. At higher blood levels, we observed a greater decline in MY and FPR and an increase in SCS. In dairy cows with a blood level > 93.7%, the reduction rates of MY, SCS, and FPR were −0.07, 0.05, and −0.02 per THI level using the BLUP method and −0.08, 0.07, and −0.05 in MY, SCS, and FPR, respectively, per THI level when using the ssGBLUP method. In contrast, the dairy cows with blood levels < 87.5% were not affected by heat stress at a THI threshold of 76.

## 4. Discussion

### 4.1. Genetic Parameter Estimation

The threshold point of heat stress in this study was higher than those previously reported. For example, this threshold ranged from 72 to 74 in US dairy cattle [24] and from 69 to 73 in dairy cattle in Belgium, Luxembourg, Slovenia, and Spain [48]. This is because almost all of Thailand’s dairy cattle are crosses of *Bos taurus* (Holstein) and *Bos indicus* (Thai native, Sahiwal, and Brahman). Bos indicus cattle can largely adapt to harsh conditions [49,50,51]. In addition, the threshold point of heat stress in this study also showed a higher threshold point than the same population in 2011 [5]. This means that this dairy herd has developed better heat tolerance than 10 years ago [5]. Both additive and permanent environmental variances from the BLUP and ssGBLUP methods were not different for variance components and heritability. Consequently, the heritability values from both the BLUP and the ssGBLUP methods were similar. These results show that the genomic and pedigree relationship matrix for the genotyped animals is similar, not altering the genetic parameter estimates. Therefore, the higher degree of relationship between pedigree and the genomic relationship could lead to more precise estimates and capture a higher additive genetic variance [35].

### 4.2. Genetic Correlation

The genetic correlations between additive effects with and without heat stress of MY ($r_{g}$) in Table 2 show low to moderate negative correlations in both the BLUP and ssGBLUP methods. These findings are similar to the results of Boonkum et al. [5], and Boonkum and Duangjinda [52]. However, the values were lower than those obtained by Aguilar et al. [4] and Bernabucci et al. [53], indicating that cows with a high milk production have a low heat tolerance. This implies that selecting animals for a high milk yield would lead to animals with greater susceptibility to heat stress. When a cow becomes heat-stressed, an immediate coping mechanism is to reduce dry matter intake, causing a decrease in the availability of nutrients used for milk synthesis [12,54]. Simultaneously, there is an increase in basal metabolism caused by the activation of the thermoregulatory system. Mild to severe heat stress can increase metabolic maintenance requirements by 7 to 25% [55], further exacerbating both the existing metabolic stress and the decrease in milk production [3]. Although the genetic correlation is small, combined selection for milk production and heat tolerance using the breeding value index is possible [25].

The genetic correlations between additive effects with and without heat stress of SCS ($r_{g}$) were positive. These results indicate that heat stress can negatively impact dairy health, e.g., causing udder inflammation, with a prevalence and incidence of subclinical and clinical mastitis, leading to low milk quality [56,57]. In contrast, the genetic correlations between additive effects with and without heat stress of FPR were negative, indicating that during heat stress, cattle reduce their metabolism, which is associated with a reduced secretion of some hormones, such as thyroid and growth hormones. Changes in FPR can be a good indicator of the energy balance, especially the development of ruminal acidosis [58]. Reduced feed intake, loss of salivary buffering from increased respiratory rates and drooling, and a reduction in the total buffering pool all contribute to a greater potential for rumen acidosis during periods of hot and humid weather [59].

In addition, the correlations of the permanent environmental effect with and without heat stress in the ssGBLUP methods were higher than the genetic correlation, and ranged from −0.339 to −0.486, which indicated that managing the environment for high MY and FPR and low SCS values would increase the negative permanent environmental effects associated with increased heat stress in cows. Thus, cows that respond to heat stress were expressed mostly by the environment rather than by genetics [4,5].

As seen in Table 3, the genetic correlation between FPR and MY was slightly negative in the BLUP (−0.21) and the ssGBLUP (−0.18) methods, suggesting that cows with a high milk production have a negative energy balance. After calving, milk production increases rapidly, reaching its maximum at 90 days in milk, whereas the increase in feed intake does not keep pace with the milk production. Consequently, the energy intake does not cover the animal’s requirements during early lactation, resulting in a negative energy balance. When combined with high temperature and humidity, the cows will reduce their feed intake, resulting in insufficient nutrition and, consequently, more severe energy imbalances [54,60]. The genetic correlations between MY vs. SCS and FPR vs. SCS were very low, indicating that these traits were not genetically correlated.

### 4.3. Accuracy

The achievement and sustainability of a breeding application incorporating genomic records largely depend on the accuracy of the predictions. The ssGBLUP technique can supply correct and much less biased genomic critiques than the BLUP approach [61,62]. The accuracy of predictions from the ssGBLUP method was better than that obtained with the BLUP method, with 32–54% for the ratio of accuracy and 60–80% for the methods. The accuracy values in this study were similar to those previously obtained [63,64]. Low accuracies can occur for a number of reasons: (1) A small reference population (number of animals with genotype data); therefore, the reference population should be greater than 1000 to significantly increase the accuracy [64]. (2) In small populations with few animal genotypes, combining phenotypes, pedigree, and genotypes for a small subpopulation and a large population may increase the accuracy of genetic predictions [62,64]. (3) Altering the use of data from young bulls to proven sires will increase the accuracy, as in the proven sires, there are data records of daughters, whereas for young bulls, there are no such records available [63]. (4) In case of low heritability traits, combination with high heritability traits in terms of multiple trait analysis can increase accuracy. In addition, the use of data from several generations is another approach [37,65]. Therefore, our study also demonstrates that the accuracies of genetic predictions can be further improved by increasing the reference population.

### 4.4. Rate of Decline of MY, SCS, and FPR

The productivity rate declined when the animal was subjected to heat stress; in other words, when the THI threshold was reached. In this study, a THI of 76 was the starting point for the Thai−Holstein crossbred population. The decline in productivity was noticeable and consistent with that observed for other dairy populations, e.g., milk and milk quality losses in Iranian Holstein [66]; US dairy [4,24]; UK dairy [67]; Belgium, Luxembourg, Slovenia, and Spain dairy [48]; in somatic cell score and somatic cell count in US Holstein and Jersey dairy [68]; Holstein dairy in Luxembourg [56]; and dairy cows in Germany [69], and in cows with a negative energy balance [70,71]. The variation in production loss depends on the breed and on the genetic merit of animals of the same breed [72]. In this study, we found smaller decreases in milk yield of around 70–80 g per point rise in THI (> 93.7% Holstein genetics), with a 20–40% lower milk loss compared to other populations. However, interestingly, milk loss in this herd, compared to the same dairy population in 2015, was reversed. In another study, the rate of decline in milk yield was twice as high in 2021 as in 2015 [52], indicating that heat stress in Thailand is intensifying.

For SCS, the rate of decline was different from that of milk yield; beyond the THI threshold, the SCS values increased [56,73]. The increase in SCS during heat stress was positively correlated with decreased immune function due to the oxidative stress, leading to udder infections and mastitis [56,74]. Moreover, Lambertz et al. [75] observed that increasing THI values increased SCS in Holsteins in four different housing systems. Godden et al. [76] stated that heat and humidity did increase the pathogen load, resulting in a significant incidence of mastitis. Negri et al. [77] reported that changes in the somatic cell score can be an early indicator of heat stress in Holstein cattle under tropical conditions, even before a decrease in milk yield.

For FPR, this is the first report of decreasing FPR values when dairy cattle are subjected to heat stress. FPR is mainly used as a diagnostic tool to determine NEB and several metabolic disorders and abomasal displacement [78]. The optimum FPR ranges between 1.2 and 1.4 for healthy cows [79]. A lower FPR (<1.2) indicates subclinical rumen acidosis, which endangers the cow’s reproductive abilities, whereas an FPR greater than 1.4 reflects an energy deficit and possible subclinical ketosis. These findings have been further confirmed by Richardt [80]; rumen acidosis is suspected when the FPR is below 1.1. Furthermore, Heuer et al. [78] found a higher risk of ketosis, displaced abomasum, ovarian cysts, and mastitis at a low FPR value. Therefore, the impact of heat stress on FPR leads to acidosis in Thai dairy cattle. In addition, in our study, for 245 days of the year, the THI was beyond 76. Thus, >93.7% of Holstein genetics, accounting for 25% of the herd, were exposed to different degrees of heat stress during 67% of the year, with a total of 819 THI units.

## 5. Conclusions

Heat stress affects the genetics of milk yield, milk quality, and dairy health. In particular, Holstein genetics > 93.7% are affected. However, Holstein genetics < 87.5% may be an option to reduce the effects of heat stress, even if such animals lack an excellent cooling system. In this study, the use of the genomic selection method demonstrated a higher accuracy of genetic parameters than the traditional method. Therefore, against the background of a warming climate, a genetic approach in terms of a genomic selection method that combines heat tolerance and economic traits should be considered.

## Figures and Tables

**Table 1 vetsci-09-00066-t001:** Description of test-day milk yield (MY), somatic cell score (SCS), and milk fat-to-protein ratio (FPR) records used for the estimation of variance components, genetic parameters, and breeding values.

Item/Traits	MY	SCS	FPR
Number of herd *x* test-month *x* test-year	24,928	24,928	24,928
Number of farm *x* calving season	218	218	218
Number of breed group x months in milk group	30	30	30
Classes of age at first calving	7	7	7
Number of animals with records	15,380	15,380	15,380
Number of animals in the pedigree	33,231	33,231	33,231
Number of animals with genotypes	882	882	882
Minimum	5	0.01	0.23
Maximum	45	10.00	3.58
Mean	14.33	3.56	1.13
Standard deviation	4.46	1.82	0.33

**Table 2 vetsci-09-00066-t002:** Genetic parameters (SE in parentheses) of test-day milk yield (MY), somatic cell score (SCS), and milk fat-to-protein ratio (FPR) records at a THI of 76 using conventional BLUP (BLUP) and single-step genome BLUP (ssGBLUP) methods in Thai−Holstein crossbred cows.

Methods	BLUP			ssGBLUP		
Parameters	MY	SCS	FPR	MY	SCS	FPR
$\sigma_{hmy}^{2}$	1.247	0.421	0.018	1.058	0.411	0.018
	(0.006)	(0.005)	(0.004)	(0.006)	(0.005)	(0.004)
$\sigma_{a}^{2}$	5.753	0.152	0.007	5.650	0.204	0.007
	(0.066)	(0.010)	(0.010)	(0.063)	(0.007)	(0.009)
$\sigma_{a_{ht}}^{2}$	0.094	0.035	0.001	0.015	0.005	0.001
	(0.011)	(0.012)	(0.012)	(0.008)	(0.009)	(0.008)
$\sigma_{a,a_{ht}}$	−0.116	0.018	−0.001	−0.107	0.014	−0.001
	(0.009)	(0.006)	(0.004)	(0.008)	(0.005)	(0.005)
$\sigma_{p}^{2}$	6.208	0.707	0.023	6.481	0.754	0.025
	(0.051)	(0.005)	(0.008)	(0.058)	(0.004)	(0.009)
$\sigma_{p_{ht}}^{2}$	0.715	0.068	0.003	0.185	0.015	0.001
	(0.006)	(0.011)	(0.011)	(0.007)	(0.008)	(0.008)
$\sigma_{p,p_{ht}}$	−0.429	0.026	−0.001	−0.532	0.036	−0.002
	(0.014)	(0.006)	(0.004)	(0.009)	(0.004)	(0.004)
$\sigma_{e}^{2}$	3.350	1.762	0.050	3.086	1.736	0.049
	(0.008)	(0.003)	(0.003)	(0.008)	(0.003)	(0.003)
$\sigma_{total}^{2}$	16.822	3.101	0.100	15.836	2.075	0.098
	(0.032)	(0.062)	(0.067)	(0.046)	(0.045)	(0.055)
$h^{2}$	0.334	0.049	0.060	0.344	0.087	0.061
	(0.003)	(0.004)	(0.006)	(0.003)	(0.003)	(0.004)
$r_{g}$	−0.158	0.247	−0.378	−0.368	0.438	−0.378
	(0.008)	(0.009)	(0.008)	(0.005)	(0.009)	(0.008)
$r_{p}$	−0.204	0.119	−0.120	−0.486	0.339	−0.400
	(0.010)	(0.008)	(0.009)	(0.009)	(0.009)	(0.008)

$\sigma_{hmy}^{2}$ = herd-month-year of test variance; $\sigma_{a}^{2}$ = additive variance of animal; $\sigma_{a_{ht}}^{2}$ = additive variance of heat tolerance effect; $\sigma_{a,a_{ht}}$ = covariance between additive variance of animal and heat tolerance effect; $\sigma_{p}^{2}$ = permanent environmental variance of animal $;\sigma_{p_{ht}}^{2}$ = permanent environmental variance of heat tolerance effect; $\sigma_{p,p_{ht}}$ = covariance between permanent environmental variance of animal and heat tolerance effect; $\sigma_{e}^{2}$ = residual variance; $\sigma_{total}^{2}$ = total variance; $h^{2}$ = heritability; $r_{g}$ = genetic correlation between additive genetic and additive genetic heat tolerance effects; $r_{p}$ = correlation between permanent environment and permanent environment heat tolerance effects.

**Table 3 vetsci-09-00066-t003:** Genetic correlation between test-day milk yield (MY), somatic cell score (SCS), and milk fat-to-protein ratio (FPR) records (above diagonal) and phenotypic correlation (below diagonal) using the BLUP and ssGBLUP methods.

Methods	BLUP			ssGBLUP		
Traits	MY	SCS	FPR	MY	SCS	FPR
**MY**	−	0.06	−0.21	−	0.09	−0.18
**SCS**	−0.10	−	0.06	−0.13	−	0.01
**FPR**	0.02	−0.01	−	0.02	0.08	−

**Table 4 vetsci-09-00066-t004:** Accuracy for estimated breeding value (EBV) and genomic breeding value (GEBV) with (slope) and without (intercept) consideration of heat tolerance at a THI of 76 in test-day milk yield (MY), somatic cell score (SCS), and milk fat-to-protein ratio (FPR) traits using the BLUP and ssGBLUP methods.

Methods	BLUP			ssGBLUP		
Traits	MY	SCS	FPR	MY	SCS	FPR
Ratio of accuracies	0.24	0.37	0.26	0.37	0.49	0.38
Accuracy	0.20	0.20	0.25	0.36	0.33	0.40

**Table 5 vetsci-09-00066-t005:** Rates of decline in test-day milk yield (MY), somatic cell score (SCS), and milk fat-to-protein ratio (FPR) at a THI of 76 separated by breed group using the BLUP and ssGBLUP methods.

Methods	BLUP			ssGBLUP		
Traits	MY (kg)	SCS (score)	FPR (%)	MY (kg)	SCS (score)	FPR (%)
Percentage of Holstein genes (percentage of the number of animals)						
BG1: < 87.5% (22%)	0.12	0.01	−0.01	0.02	0.00	0.00
BG2: 87.5 to 93.6% (53%)	−0.01	0.02	−0.01	−0.03	0.04	−0.01
BG3: >93.7% (25%)	−0.07	0.05	−0.02	−0.08	0.07	−0.05

## Data Availability

The data are available upon request from the corresponding author.

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
