# Peer review of "Determining Heat Stress Effects of Multiple Genetic Traits in Tropical Dairy Cattle Using Single-Step Genomic BLUP"

_vetsci, 2022, doi:10.3390/vetsci9020066_

Round 1

Reviewer 1 Report

According to the authors, the THI is one of the most used indices in studies on thermal stress. However, it does not mean that it is the most suitable. Several studies show that the BGTHI (an index that includes the wind effect) is more accurate than the THI when used to determine heat stress levels in animals. In future studies, this could be an index to consider.

Author Response

To Reviewer,

Thank you very much for your suggestion which will help us improve our research on heat stress in livestock animals in tropical regions like Thailand. We agree that there are still many interesting THI indices to study. Therefore, this will be one of the research questions we will study in our further studies.

Sincerely yours

Wuttigrai Boonkum

Reviewer 2 Report

This paper reports the effect of heat stress manifestation on genetics for milk yield, milk quality, and dairy health traits with and without genomic information using best linear unbiased prediction (BLUP) and single-step genomic (ssGBLUP) in Thai-Holstein crossbred.  There are some very interesting findings (or confirmation) of genomic selection and methods for genetic evaluation in this report. The major conclusion is that ssBLUP technique can supply correct and much less biased genomic critiques than the BLUP approach. The paper also concluded that the genomic selection method has a higher accuracy of genetic parameters than the traditional method. Overall I consider that this manuscript would warrant publication in veterinary Sciences.

Author Response

To Reviewer,

Thank you very much for your comments and for sacrificing your valuable time to carefully review our research. Your feedback supports and encourages us to develop more research to solve livestock problems in the future.

Sincerely yours

Wuttigrai Boonkum

Reviewer 3 Report

  1. The table 1 in lines 109-111 and lines 186-188 is repeated.
  2. The reference [43] for my opinion seems as has the same interpretation of article (objectives, aim) but methods that was used in this manuscript and the article (J.Dairy Sci.) are different: ssGBLUP and ssGREML. Dataset is also the same.
  3. In table 4 we can find values for accuracy of EBVs. For SCS and FPR traits revealed by BLUP and ssGBLUP approaches it is 0.20-0.25 and 0.33-0.40 respectively. At the same time heritability for these traits was 0.049-0.087 for SCS and 0.060-0.061 for FPR which is an order of magnitude lower than for MY (0.334-0.344). But the accuracy of MY was particularly the same level compared to other traits. How can you explane this discrepancy?
  4. Main question: Why did you study only milk production features of animals? I would suggesting to add fertility traits in the article.

Author Response

To Reviewer,

We have revised the manuscript and answered questions and suggestions. Please see the details of the answers below and the attached file.

Point 1: The table 1 in lines 109-111 and lines 186-188 is repeated.

Response 1: Thank you for this point, we already deleted table 1 in lines 195-198 from our manuscript.

Point 2: The reference [43] for my opinion seems as has the same interpretation of article (objectives, aim) but methods that was used in this manuscript and the article (J.Dairy Sci.) are different: ssGBLUP and ssGREML. Dataset is also the same.

Response 2: Even the dataset is similar, the objectives of both studies are different. The former study needs to compare the effect of heat stress over time (between ten years ago and the present) on genetic parameters of milk yield using ssGREML. Meanwhile, the present study needs to prove that the new method (ssGBLUP) could be used as a potential method for the dairy breeding program of Thailand instead of the traditional BLUP method. Besides milk yield, which is only one parameter in the former article, FPR and SCS traits were analyzed in the present study. Those traits were parameters for milk production and infer to dairy health and milk quality in the large data.    

Point 3: In table 4 we can find values for accuracy of EBVs. For SCS and FPR traits revealed by BLUP and ssGBLUP approaches it is 0.20-0.25 and 0.33-0.40 respectively. At the same time heritability for these traits  was 0.049-0.087 for SCS and 0.060-0.061 for FPR which is an order of magnitude lower than for MY (0.334-0.344). But the accuracy of MY was particularly the same level compared to other traits. How can you explane this discrepancy?

Response 3: According to the above mention, the heritability and the accuracy were calculated by different equations; therefore, those were not completely related each other. Genes control the heritability of the traits. High heritability means phenotypes depend on genetics more than genotypes. Accuracy is the mathematic equation to compare the genetic methods by several equations. The different accuracy values result from many factors, such as the number of datasets and data structures, the heritability of each trait, and methods of analysis. Therefore, those are not entirely correlated.

Point 4: Main question: Why did you study only milk production features of animals? I would suggesting to add fertility traits in the article.

Response 4: Milk production is an economically major trait affecting benefits to dairy farmers. Therefore, improving milk production traits directly brings more significant dairy farming profitability. Also, high somatic cells are related to mastitis, a major problem in the Thai dairy population because that results in decreased milk synthesis [26]. Besides, milk fat to protein ratio is an easily measurable trait and can be used as an indicator to evaluate negative energy balance (NEB) ketosis, which is related to decreasing milk yield [27]. Therefore SCS and NEB were indirect indicators for dairy health and had the impact to evaluate in terms of large data. For the reproductive traits, it is interesting; however, those are quite difficult to study together as data characteristics are different. It would be better to study separately. Therefore, we are studying the reproductive traits in crossbred Holstein and hope to publish in the future.   

For more clarify in our objective, we therefore added the above mention in the introduction. See lines 61-71.

References:

  1. Buaban, S.; Lengnudum, K.; Boonkum, W.; Phakdeedindan, P. Genome-wide association study on milk production and somatic cell score for Thai dairy cattle using weighted single-step approach with random regression test-day model. Dairy Sci. 2020, 105, 468-494.
  2. Schcolnik, T. Using milk fat-to-protein ratio to evaluate dairy cows energy balance status. J. Anim Sci. 2016, 94, 54-55.

Sincerely yours

Wuttigrai Boonkum
